# Stability of Non-Ionic Surfactant Vesicles Loaded with Rifamycin S

**DOI:** 10.3390/pharmaceutics14122626

**Published:** 2022-11-28

**Authors:** Verdiana Marchianò, Maria Matos, Ismael Marcet, Maria Paz Cabal, Gemma Gutiérrez, Maria Carmen Blanco-López

**Affiliations:** 1Department of Chemical and Environmental Engineering, University of Oviedo, Julián Clavería 8, 33006 Oviedo, Spain; 2Department of Physical and Analytical Chemistry, University of Oviedo, Julián Clavería 8, 33006 Oviedo, Spain; 3Department of Organic and Inorganic Chemistry, Instituto de Química Organometálica “Enrique Moles”, University of Oviedo, Julián Clavería 8, 33006 Oviedo, Spain

**Keywords:** niosomes, synthesis and characterization, drug delivery, stability, antimicrobial activity

## Abstract

These days, the eradication of bacterial infections is more difficult due to the mechanism of resistance that bacteria have developed towards traditional antibiotics. One of the medical strategies used against bacteria is the therapy with drug delivery systems. Non-ionic vesicles are nanomaterials with good characteristics for encapsulating drugs, due to their bioavailability and biodegradability, which allow the drugs to reach the specific target and reduce their side effects. In this work, the antibiotic Rifamycin S was encapsulated. The rifamycin antibiotics family has been widely used against *Mycobacterium tuberculosis*, but recent studies have also shown that rifamycin S and rifampicin derivatives have bactericidal activity against *Staphylococcus epidermidis* and *Staphylococcus aureus*. In this work, a strain of *S. aureus* was selected to study the antimicrobial activity through Minimum Inhibitory Concentration (MIC) assay. Three formulations of niosomes were prepared using the thin film hydration method by varying the composition of the aqueous phase, which included MilliQ water, glycerol solution, or PEG400 solution. Niosomes with a rifamycin S concentration of 0.13 μg/g were satisfactorily prepared. Nanovesicles with larger size and higher encapsulation efficiency (EE) were obtained when using glycerol and PEG400 in the aqueous media. Our results showed that niosomes consisting of an aqueous glycerol solution have higher stability and EE across a diversity of temperatures and pHs, and a lower MIC of rifamycin S against *S. aureus*.

## 1. Introduction

In recent years, the use of nanocarriers has been extensively studied and exploited in various fields, such as the pharmaceutical, cosmetic, and food industries. They have numerous advantages, including the ability to transport a certain drug to its specific target, loading either hydrophilic or hydrophobic drugs, increasing drug pharmacokinetics, and decreasing possible side effects [1]. Niosomes are vesicles that can act as nanocarriers, similar to liposomes, which have a structure formed by a membrane bilayer surrounding an aqueous environment. Membrane components are non-ionic surfactants such as sorbitan derivatives or polysorbates with cholesterol as a stabilizing agent [2]. Unlike liposomes, niosomes have a greater physical–chemical stability, and present more versatility since they can be produced on a large scale with lower costs for the materials used, and also have better environmental compatibility and biodegradability. The methods of preparation are the same as for liposomes. In this work, the method used is the thin film hydration method, commonly used to obtain stable large multilamellar vesicles [3].

These nanomaterials are widely used in the therapeutic field for the transport of anticancer and antimicrobial drugs [4,5,6].The excessive use of conventional antibiotics, and other factors, have contributed to the steady rise of Antimicrobial Resistance (AMR) among many pathogenic bacteria. The drug delivery approach is a way to try to overcome this problem by allowing the continued use of traditional drugs against the pathogenic bacteria, by camouflaging the antibiotic and bringing them directly to the site of action [7].

One of the most aggressive bacteria causing severe infections that are difficult to eradicate due to multi-drug resistance is *S. aureus*. Rifamycins are a family of antibiotics with a similar naphthalene ansamycin framework, mechanism of action, and therapeutic efficacy. The rifamycin are generally active against Gram-positive bacteria, especially *Mycobacterium tuberculosis*. Their mechanism of action is based on the inhibition of DNA-dependent RNA polymerase, which leads to the blocking of bacterial RNA synthesis inside the microorganism [8]. In recent studies, their bactericidal activity against *S. Aureus* and its biofilms has been confirmed [9,10,11].

In the present study, rifamycin S was encapsulated in niosomes prepared with Span^®^ 60 and cholesterol as membrane constituents, which were chosen due to their stable structure and high entrapment capacity [12]. To examine possible synergistic effects on the rifamycin’s antimicrobial activity [13], one of three different aqueous phases was examined: pure MilliQ water, a mixture of MilliQ water and glycerol (60:40 *v*/*v*), and a mixture of MilliQ water and PEG 400 (55.3:44.7 *v*/*v*). These three formulations were studied to examine changes in vesical morphology, surface charge, size and Encapsulation Efficiency (EE) [14]. The antimicrobial activity of these rifamycin-encapsulated niosomes was assessed by determining the Minimum Inhibitory Concentration (MIC) values. In addition, Multiple Light Scattering (MLS) experiments were carried out by varying two fundamental parameters known to affect vesical structure and stability, temperature, and pH.

## 2. Materials and Methods

### 2.1. Chemicals and Reagents

Surfactants used for the membrane components of the niosomes were Span^®^ 60 (Sigma-Aldrich, Sant Louise, MO, USA) and cholesterol stabilized at 96% (Acros Organics, Geel, Germany), each dissolved in absolute ethanol (J.T. Baker, Avantor, Bridgewater, NJ USA). The hydration phases used during the preparation method were polyethylene glycol 400 (PEG-400, MW 380−420 g/mol, density 1.128 g/cm^3^, VWR International, Barcelona, Spain), glycerol bi-distilled 99.5% (GLY, MW 92.09 g/mol, density 1.261 g/cm^3^, VWR International LLC, BDH PROLABO) and ultrapure MilliQ water. Rifamycin S (MW = 697.78) was procured from the Department of Organic and Inorganic Chemistry, University of Oviedo. The *S. aureus* strain used for the MIC assay was generously donated by the Dairy Research Institute of Asturias (Instituto de Productos Lácteos de Asturias, IPLA-CSIC, Villaviciosa), Asturias, Spain.

### 2.2. Preparation of Vesicles

Vesicles were synthesized using a standard thin film hydration method, with some modifications to make it suitable at a small scale. Classical thin film hydration consists of dissolving the surfactants in the organic phase and then evaporating using a rotary evaporator at reduced pressure. When the film is formed, it is hydrated with the aqueous phase by rotating at a certain temperature, facilitating the formation of vesicles. In the modified method we used, instead of using the rotary evaporator, a stream of nitrogen was used to slowly evaporate the organic phase to create the dried film. The hydration was made by adding the aqueous phase directly to the dried film formed, and sonicating (Branson Ultrasonics Sonifier SFX150, Tamaulipas, Mexico) for 10 min using an amplitude of 45%, 500 W power, and 20 kHz frequency. The procedure is summarized in Figure 1. Membrane components used were Span^®^ 60 and cholesterol in a fixed 2:1 molar ratio, with a total concentration of 8 g/L in 800 µL of absolute ethanol as the organic phase and 2 mL of MilliQ water as the aqueous phase. These parameters were chosen after several preliminary tests using different volume ratios of organic and aqueous phases.

After optimizing the formulation and preparation procedure to create empty vesicles, the procedure was repeated to encapsulate rifamycin S. For this purpose, a stock solution of 1 mg/mL of rifamycin S in absolute ethanol was prepared. Then, 20 µL of this stock solution were added into the organic phase where the membrane components were dissolved. Due to rifamycin’s hydrophobic nature, and to provide further hydration, one of three different aqueous media was then added: MilliQ, MilliQ: glycerol (60:40 *v*/*v*), MilliQ: PEG400 (55.7:44.3 *v*/*v*).

### 2.3. Vesicles Purification

Ultracentrifugation and filtration were performed to remove impurities, non-encapsulated antibiotic, extraneous membrane components, and empty vesicles. Ultra 0.5 mL centrifugal filter devices were used in a microcentrifuge (Espresso centrifuge, Thermo electron Corporation, Waltham, MA, USA) for 15 min at a 14.500 rcf. The supernatant restored with pure aqueous phases was filtered using PES syringe filters with a 0.22 µm pore diameter and then analyzed by Reverse Phase High Performance Liquid Chromatography (RP-HPLC).

### 2.4. Vesicles Size Distribution and Morphology

The average hydrodynamic sizes and zeta potential values of the synthesized vesicles were measured by Dynamic Light Scattering (DLS) on a Zetasizer NanoZS series (Malvern Instruments Ltd., Malvern, UK).

Negative Staining Transmission Electron Microscopy (NS-TEM) was used to study vesicle morphology with a JEOL-2000 Ex II transmission electron microscope (Tokyo, Japan). The negative staining of the sample was obtained by dropwise addition of 2% (*w*/*w*) phosphotungstic acid solution to the copper carbon-coated grill.

### 2.5. Stability

Turbiscan Lab Expert (Formulaction Co., Toulouse, France) equipment was used to determine the formulation’s stability under different conditions of pH and temperature. For this purpose, samples were diluted 1:10 with deionized water, and 20 mL of this diluted sample were placed in glass tubes. The measurements were carried out by Static Multiple Light Scattering (S-MLS). The incident beam used was a near infrared light with a wavelength of 880 nm. The backscattering (BS) was reported along the cell height. The samples were analyzed every 24 h at 30 °C, 40 °C, and 60 °C for 3 days and at room temperature with a pH 2, 7, and 9.

### 2.6. Encapsulation Efficiency (EE)

The calculation of encapsulation efficiency (EE) gives information about the amount of antibiotic inside the vesicles. To perform this analysis, purified and non-purified vesicles were diluted using methanol 1:10 (*v*/*v*), necessary to break the membrane bilayer and release the encapsulated compound.

EE was calculated according to Equation (1):(1)$$EE\%=\frac{\left[Drug\;concentration\;in\;purified\;vesicles\right]}{\left[Drug\;concentration\;in\;no\;purified\;vesicles\right]}\times 100$$

The measurements were performed by RP-HPLC (HP series 1100 chromatograph, Hewlett Packard, Agilent Technologies), using a Zorbax Eclipse Plus C18 column (4.6 mm × 150 mm, 5 μm, Agilent Technologies, Santa Clara, CA, USA). The chromatographic method used included a linear gradient with MilliQ water (mobile phase A) and acetonitrile (mobile phase B). The gradient started with 20% of B, obtaining 100% of B at 5 min and kept constant for 10 min. The flow rate was 0.8 mL/min. Retention time for rifamycin S was 6.7 min at λ = 254 nm.

The load capacity (LC) of the vesicles was also calculated to quantify the amount of drug encapsulated. Equation (2) was used:LC = Wb/WT(2)

Wb corresponds to the mass (mg) of the drug added and WT is the total amount of drug and membrane components of the vesicles.

### 2.7. Minimum Inhibitory Concentration (MIC) Assay

The Minimum Inhibitory Concentration (MIC) is defined as the minimum concentration of a compound that is able to inhibit the growth of bacteria. MIC was determined by visual inspection of the bacteria deposition at the bottom of the inoculated wells in a 96-well plate. In this case, the incubation time was 24 h at 30 °C and the concentrations of antibiotic tested for every sample (free antibiotic or total antibiotic concentration in the encapsulated samples) were 1 µg/mL, 0.75 µg/mL, 0.5 µg/mL, 0.25 µg/mL, 0.1 µg/mL, 0.05 µg/mL, 0.025 µg/mL.

*S. aureus* was grown overnight in a liquid medium (Tryptic Soy Broth, TSB) at 37 °C by transferring a single colony from a nutrient agar plate into 150 mL of TSB. The concentration of the bacteria was measured by OD600 and then an aliquot of this growth medium was diluted to 1 × 10^5^ CFU/mL in TSB. Afterwards, a volume of 50 µL of this concentration of bacteria was inoculated into each well of a sterile 96-well plate containing 50 µL of free or encapsulated antibiotic at the concentrations described above. A positive control sample containing only the test bacteria and a negative control with only with only vesicles were also included in the assays.

### 2.8. Statistical Analysis

All data were expressed as the mean ± SD (standard deviation) of three independent experiments, and statistical analysis of the data was carried out (ANOVA). Fisher’s test (*p* < 0.05) was used to calculate the least significance difference (LSD) using Microsoft Excel.

## 3. Results and Discussion

### 3.1. Characterization: Size and Morphology

Size distribution and morphology of nanovesicles were characterized in terms of mean particle size and zeta potential. The results showed in Table 1 underline how the aqueous phases can influence the size of vesicles, confirmed by previous works [14,15]. Using pure MilliQ water as aqueous media led to vesicles with a small size, around 60 nm, but with higher Z-potential, which indicates higher electrostatic stability compared to the other two aqueous media tested. The size increases when using MilliQ:glycerol (60:40 *v*/*v*) solution and it gets even bigger with MilliQ-PEG 400 (55.7:44.3 *v*/*v*), with size values of around 345 nm and 1500 nm, respectively. Data were confirmed by size distribution in Figure 2. The Z-potential absolute value in these two systems is very low, with a positive charge in the case of the glycerol solution and a negative charge with the PEG400 solution due to the presence of a carboxylic group [16].

Z-potential is commonly used as an index of stability of the colloidal suspensions under electrostatic repulsion. The suspension is generally considered stable with absolute values larger than 30 mV. If the values are very low, phenomena such as flocculation or coalescence, and hence creaming or phase separation, could be possible [17].

The Span^®^ 60 molecule has a long alkyl chain, which reduces the interaction between the polar heads of the amphiphilic molecules, thus being a suitable surfactant for stable niosomes formation [18]. Cholesterol is a hydrophobic molecule commonly used as a stabilizer in vesicle formulation. Its use can increase vesicle size, as well as stability and EE [12]. However, in the present work it was observed that the use of glycerol and PEG400 as hydration media produced interactions with vesicle membrane compounds that led to an increase on the final vesicle size, which was especially noticeable when the PEG400 solution was used. Similar results were reported in other vesicles formulations when the same hydration media were used [15]. Glycerol has been used in other work to obtain vesicles of larger size, as it increases the radius of the curvature of the membrane, allowing the formation of multilamellar vesicles with greater fluidity [19]. The use of PEG400 contributes to the formation of vesicles through hydrogen bonding and hydrophilic action; its concentration is important because an excess of PEG can lead to a decrease in membrane rigidity and the stability of the niosomes [20]. Nevertheless, due to its structure, PEG occupies more space in the vesicle’s core, making the vesicles larger. It is also important to take into account that the nature of the molecule encapsulated could also have an important effect on the final vesicle size [15,21].

Morphology of vesicles was examined by HR-TEM; the larger size of vesicles in glycerol and in PEG400 are confirmed in Figure 3.

According to the results, the EE is affected by the use of different aqueous media (Figure 4a). In general, the EE values obtained were high for the three formulations tested, considering that the EE is dependent on the size and properties of the encapsulated molecule, and its interaction with the membrane double-layer and perhaps other components of the niosomal vesicles. Indeed, Span^®^ 60 is a surfactant, having a small polar head and greasy long alkyl chain [22], which enables enhanced entrapment of the relatively lipophilic rifamycin S. The EE values and the vesicles sizes follow the same trend, with the highest EE values obtained when the PEG400 solution was used as hydration media, and the lowest when pure MilliQ was used. In other studies, PEG has been found to improve the EE and drug solubility of lipophilic molecules It has also been reported that glycerol has a similar effect, at least up to 45% of the aqueous composition, while a larger concentration of glycerol could decrease the final EE [23].

The EE calculation was made according to Equation (1), registering values of 76% for vesicles hydrated with pure MilliQ, 81% for those hydrated with glycerol solutions and for 84% for those hydrated with PEG400 solution, as illustrated in Figure 4a.

While noting the differences in the observed EE values, it is important to point out that the calculated LC values were all similar for the three systems studied, albeit being slightly larger in the cases that glycerol or PEG solutions were used as the hydration media (Figure 4b).

### 3.2. Antimicrobial Activity of Niosomes: MIC Assay

The formulations that presented higher EE and size were chosen to test their antimicrobial activity through the MIC assay, which were those in which glycerol and PEG400 solutions were used as hydration media. MIC concentrations obtained are shown in Table 2. Even considering the turbidity of the niosomes solutions, it was possible to see the accumulation of the bacterial cells at the bottom of the wells.

As expected, no bacterial growth was observed for the negative control, and the *S. aureus* was able to grow adequately in the liquid medium in the absence of the antibiotic (control positive).

The Rifamycin S-free solution presents a MIC lower than the tested concentrations (<0.025 µg/mL), while the encapsulated rifamycin S samples afforded higher MIC values, 0.025 and 0.05 µg/mL for niosomes containing glycerol and PEG solutions, respectively. This indicates that the antibacterial effect of encapsulated antibiotic is lower than that of the free antiobiotic. However, as aforementioned, encapsulation could reduce its side effects and control its delivery.

In order to study the possible antimicrobial effect of the niosomal systems, the same dilutions were prepared for empty systems (niosomes without antibiotic).

According to these results, niosomes prepared with PEG solutions but without antibiotic were unable to produce any inhibitory effect on S. aureus, but the same vesicles loaded with the antibiotic showed a noticeable antimicrobial activity. On the other hand, empty niosomes in which a glycerol solution was used as hydration media showed antimicrobial activity. This could be attributed to the presence of glycerol, since glycerol has been described as an antibacterial agent by other authors [24]. The minimum glycerol concentration required in empty vesicles to present antibacterial activity was 50.4 µg/mL.

In order to assess the possible synergistic effect of rifamycin S and glycerol, a drug combination analysis (isobole method) described by Tallarida [25] was carried out. The MIC obtained for rifamycin S in niosomes containing glycerol was 0.025 µg/mL, which corresponds to a glycerol concentration of 2.52 µg/mL. To study the antibacterial effect of both compounds separately, empty vesicles with glycerol were used to evaluate the effect of glycerol alone, while to study the effect of the encapsulated rifamycin S, niosomes with PEG were used, since PEG was found to not exert an antibacterial effect. The results indicate that the combination of both compounds had a subadditive effect, since the concentration obtained was located above the isobole line described by 50% of the inhibitory concentration value (MIC_50_) found for each individual compound.

### 3.3. Effects on Niosomes Stability by Altering Physicochemical Parameters

In the second part of the present work, temperature and pH were modified to investigate their effects on vesicle features characterization, and to evaluate its stability under different conditions. Parameters such as size, morphology, stability, and EE of the three formulations were studied. Experiments were carried out at temperatures higher than room temperature, 30 °C, 40 °C, and 60 °C, and in aqueous media that varied from acidic (pH = 2), to neutral (pH = 7), to basic (pH = 9).

#### 3.3.1. Influence of Temperature

Temperature can modify the size, and consequently the surface charge, of vesicles, because of the structure of the vesicles in which a bilayer is formed. This is dependent on the type of surfactant used, and vesicle formation occurs when the surfactants are in a gel–liquid phase transition temperature (TC), in which the alkyl chains are well-organized. TC affects the EE. In this work, Span^®^ 60 is used for its higher TC to enhance the EE [26]. The increase in temperature leads to a thermal shock that can induce changes in vesicle size, shape, or even cause the vesicle to rupture. By making the membrane of the vesicles more fluid, it is also possible that more aqueous phase is incorporated within the vesicle core, and hence EE could suffer variations [27].

In Figure 5 it can be observed that niosomes with encapsulated rifamycin S hydrated with pure MQ water increases in size when warmed up from 30 to 40 °C. However, differences were not observed when they were warmed further to 60 °C, probably because it is near to the value of the phase transition temperature (TC) of Span^®^ 60. After 3 days of storage at the mentioned temperatures, the vesicles increased in size, especially those storage at 40 and 60 °C, indicating that temperature could contribute to molecular relaxation, thus increasing surface curvature radius and hence vesicle size. This trend was also observed for the morphology among freshly-prepared samples, and after 3 days of storage. Figure 6 shows images obtained from NR-TEM.

Table 3 summarizes the values of mean size, zeta potential, EE, and Turbiscan Stability Index (TSI) of freshly-made samples and those after being heated at 30, 40, and 60 °C, and after 3 days of storage at these same elevated temperatures.

It could be observed that the zeta potential values did not suffer large differences for niosomes hydrated with pure MilliQ at any temperature, being large negative values in all cases. Only a decrease in the zeta potential value was observed at the highest temperature tested (60 °C) after three days of storage.

Regarding vesicle stability, the parameters considered in our study were the backscattering (BS) and the TSI. TSI allows samples to be compared according to an overall stability with a global numerical number, where large TSI values indicate greater instability [28,29]. BS profiles are shown in Appendix A. From these BS profiles we it can be observed that samples hydrated with pure MilliQ did not present creaming or sedimentation, and, in all cases, changes in size were registered according to the differences observed on BS light along the cell. No large differences were observed for the three temperatures tested, but the TSI values indicated higher stability for the vesicles stored at 60 °C.

Niosomes prepared using glycerol solution as the hydration media increased in size when heated from 30, to 40, and to 60 °C. However, it was observed that after three days, storage at these elevated temperatures caused the niosomes size to be reduced. The smallest sizes were seen at 60 °C, while the samples stored at 30 °C were more similar to the freshly-prepared ones and those stored at 40 °C.

Differences in the zeta potential values were found, in that fresh vesicles at room temperature had zeta potential values close to neutral, indicating that electrostatic repulsion at the surface of the vesicles was not taking place, as was expected considering that non-ionic surfactants were used for the formulations. However, an increase in temperature produced a positive Z-potential value (17–44 mV) for the heated vesicles. However, after 3 days of storage at the elevated temperatures, the Z-potential of the niosomes returned close to zero. This effect may indicate that changes in temperature could affect molecular organization on the membrane bilayer, affecting the surface charge, but after a period of time, the molecules could relax to their original organization.

For niosomes hydrated with glycerol solution, the greatest stability was also observed for those samples heated to 60 °C, while larger changes in BS was observed at lower temperature (Appendix A. This was also indicated by the measured TSI values (Table 3). Moreover, as a general trend, the vesicles prepared in glycerol solution showed higher stability than the ones hydrated with pure MilliQ.

Niosomes hydrated with the PEG400 solution had a larger initial size compared to the other two solvent systems tested. For niosomes heated at 30 °C, similar behavior was observed to those formulated with glycerol solution, and this remained constant after three days. No changes in the Z-potential values were observed for these samples, with all being close to zero. The exception was for the samples stored for 3 days at 60 °C, in which case we observed an increase in the Z-potential value up to nearly 35 mV.

In the case of niosomes hydrated with pure MilliQ and glycerol solution, Turbiscan profiles indicated that niosomes stored at higher temperature increased their stability. Those stored at 60 °C were the most stable, with lower TSI values. However, collectively, their stability was lower than that of niosomes prepared with glycerol solution.

Table 3 shows the results obtained for the samples after three days at the indicated temperatures, versus those freshly made. In the case of niosomes formulated using MilliQ water, no significant variation in EE was observed either at 30 °C or 40 °C. However, vesicles kept at 60 °C showed a decrease in EE, indicating the possible leakage of encapsulated antibiotic.

A different trend was observed for niosomes formulated in the glycerol solution. For these, the EE remained constant until 40 °C, but when they were stored at 60 °C, an increase in EE was observed, reaching values of up to 96%. It is important to point out that this formulation was the most stable when using Turbiscan. Therefore, it is clear that the increase in temperature affects the stability of the double layer, entrapping a larger amount of rifamycin S [30]. Niosomes prepared in PEG400 presented the highest EE when freshly-prepared, but the encapsulated rifamycin S concentration decreased when temperature was increased. The EE values were uniformly lower compared to the ones found when pure MilliQ and glycerol solution were used as the hydration media, at the three temperatures tested.

Taking into account the LC of all systems, it can be observed that fresh vesicles prepared with MilliQ had lower values. However, for samples stored during three days at different temperatures, those prepared with the PEG solution gave lower LC values. As a general trend, niosomes prepared with glycerol present the highest LC values, which remains constant over time regardless of storage temperature.

#### 3.3.2. Influence of pH

Variations of pH in the environment of the samples can affect the properties and behavior of the particles and the dispersion media, especially their surface charge. Other consequences could be an osmotic shock that causes deformation of vesicles, larger sizes or agglomeration, and even rupture of the vesicles [31]. At low pH, the semipermeable structure of the membrane is influenced by the addition of hydrochloric acid, due to the higher concentration of H+ ions in which the bilayer and PEG polymers are impermeable. This provokes an osmotic pressure that pushes the water out of the vesicles and, consequently, causes their deformation in shape, or, in extreme cases, membrane disruption [32]. Finally, it is also important to take into account that if the drug encapsulated becomes a charged molecule, the increase in volume of the aqueous phase would likely separate the membrane bilayer due to charge repulsion, resulting in increased size of the vesicles [33]. It is also important to consider that the physical stability of the niosomes is influenced not only by the electrostatic forces, but also by steric interactions [34].

All data obtained is summarized in Table 4 for assessment of the niosomes at the three pH values. Moreover, Figure 7 shows particle size distributions, and Figure 8 shows the HR-TEM pictures obtained.

As mentioned earlier, niosomes in which pure MilliQ water was used as the hydration media had a mean size of around 55 nm, with a zeta potential value of −31 mV, when the pH of MilliQ water was 5.5. By changing the pH, it is possible to see a remarkable size variation dependent on pH. Indeed, at pH 2, vesicles reached a size around 340 nm and a positive Z-potential (~14 mV). This can be explained by considering the interactions between the membrane components, Span60 and cholesterol, and the acidic environment with a higher presence of H+ ions [35]. At pH 7 and 9, the mean size registered was around 165 nm, being still larger compared to the initial one at lower pH. In this case, the Z-potential values were slightly higher in both cases, but still negative with values around −40 mV, which indicate increased stability of vesicles. However, values close to zero were observed at pH 2, probably due to the presence of ions H+ that neutralized the anionic surface charge. The measurements were repeated after 3 days under storage at the same pH. However, no large differences were found in the mean size and zeta potential values of the vesicles

Niosomes hydrated using the glycerol solution were larger in size compared those prepared using MilliQ. In a similar trend to the one obtained with niosomes synthesized in MilliQ water, there was a decrease in size going from the acidic pH to the basic one, as can be seen in Table 4. Z-potential did not suffer large differences when niosomes were treated at several pHs, all being low positive values, indicating that vesicle stability was more related to steric forces than electrostatic ones. After three days of storage at these same pHs, there was slight change in size in all cases and the zeta potential values remained close to zero in all cases.

Niosomes hydrated with the PEG400 solution showed an opposite behavior from the other two formulations tested, with the mean size increasing when going from acidic (pH 2) to basic (pH 9) media. At pH 2, the mean particle size was 842 nm, which is smaller than the size of the vesicles at pH 7, where the size increased to around 1500 nm. At pH 9, the mean particle size grew further, to 1745 nm. At pH 7, the niosomes were very stable, with Z-potential values of around 33 mV. The value did not change much, staying at around 20 mV, at the other two pHs tested. After three days of storage, both size and Z-potential values decreased, indicating diminished stability of the PEG400 samples. These changes were confirmed in the stability analysis in which large changes in BS profiles and TSI values can be observed. In other studies, it was found that polymers had interactions with drugs through electrostatic bonds, van der Waals forces, and hydration bridges, which could be modified at different pHs [36].

As a general trend, it was observed that basic conditions produced more stable vesicles with lower TSI values. The vesicles hydrated with PEG400 solution as hydration media show lower stability.

Regarding the results shown in Table 4, in the formulations made with pure MilliQ water there was a decrease in the EE value at pH 2 (61%) compared to the original fresh sample (76%), but an increase at pH 7 (86%) and pH 9 (91%), so there is a loss of compound from the vesicles in the acidic media. Niosomes formulated with glycerol at the extreme pHs of 2 and 9 had lower EE values (63% and 70% respectively) than the ones measured for the fresh, original sample (81%), while at pH 7 the EE increases by 10% to a value of 91%. In the case of niosomes hydrated with the PEG400 solution, at acidic pH we see a large reduction on the EE values (from 84% to 49%) while at pH 7 the EE value remained at 85%, and at pH 9 there was an increase to 93%.

Regarding the LC of the systems studied, at acidic pH vesicles formed with the PEG solution had higher values. After three days of storage at neutral conditions, all systems present similar LC values, which were similar to those for fresh vesicles. Nevertheless, at basic pH those vesicles in which glycerol was used for its formulation present lower LC than those prepared with the MilliQ or PEG solutions.

## 4. Conclusions

Rifamycin S was encapsulated in niosomes with high EE using different formulations, by changing the aqueous phases used for the synthesis from pure MilliQ water to glycerol and PEG400. It was shown that size and EE increased with the use of both. Niosomes containing the glycerol solution as a hydration media had more potent antibacterial activity towards MRSA, as confirmed by an MIC assay.

The formulations were tested also at higher temperatures and at acidic, basic and neutral pHs. Niosomes hydrated with pure MilliQ water showed lower EE values of rifamycin S encapsulation, but its resistance to the preservation under different conditions were constant at the different temperatures tested. Niosomes formulated using PEG400 gave larger EE values but, this decreased when they were heated at different temperatures. On the other hand, niosomes hydrated with glycerol solution presented the highest values of EE at all temperatures tested. As a general trend, higher stability was found when niosomes were stored at high temperatures, at basic pH, and prepared using glycerol solution.

The three types of formulations tested showed greater stability and higher EE values when they were preserved at basic pH, with values of drug loading efficiencies (EE) up to 90%, giving niosomes have a concentration of up to 0.13 μg of rifamycin S per gram.

## Figures and Tables

**Figure 1 pharmaceutics-14-02626-f001:**
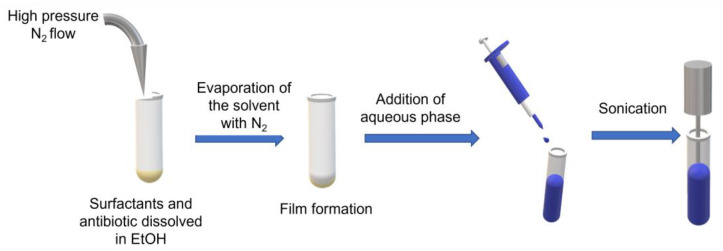
Thin film hydration method at small scale.

**Figure 2 pharmaceutics-14-02626-f002:**
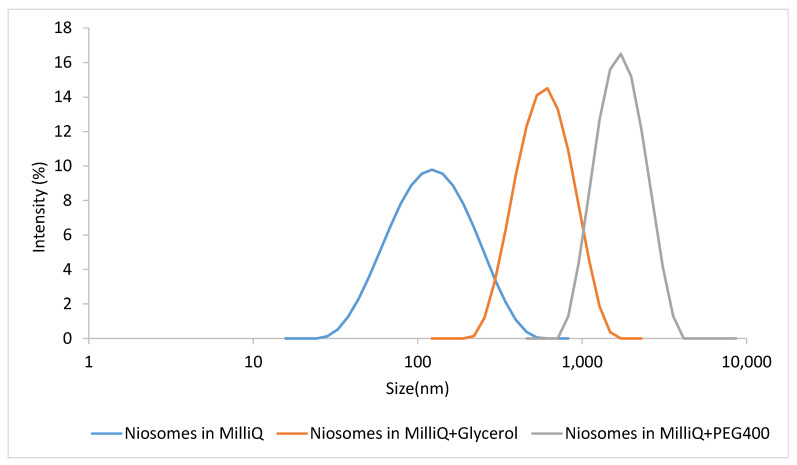
Size distribution of niosomes referring to the intensity, in MilliQ water, in mixture of MilliQ water and glycerol and in mixture of MilliQ water and PEG400.

**Figure 3 pharmaceutics-14-02626-f003:**
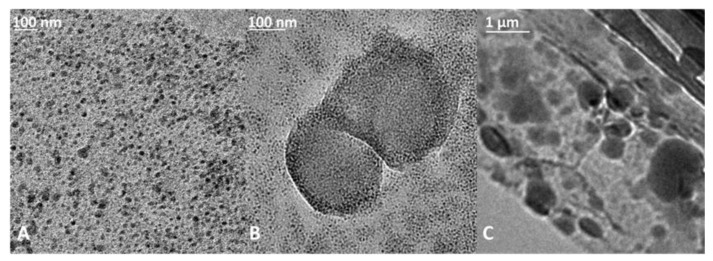
Images from HR-TEM of: (**A**) niosomes in MilliQ water; (**B**) niosomes in MilliQ and glycerol; (**C**) niosomes in MilliQ and PEG400.

**Figure 4 pharmaceutics-14-02626-f004:**
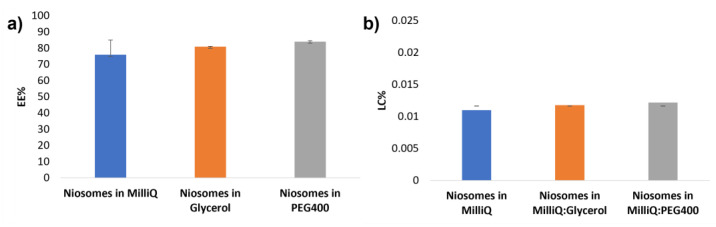
EE (**a**) and LC (**b**) values for the different niosomes formulations of rifamycin S.

**Figure 5 pharmaceutics-14-02626-f005:**
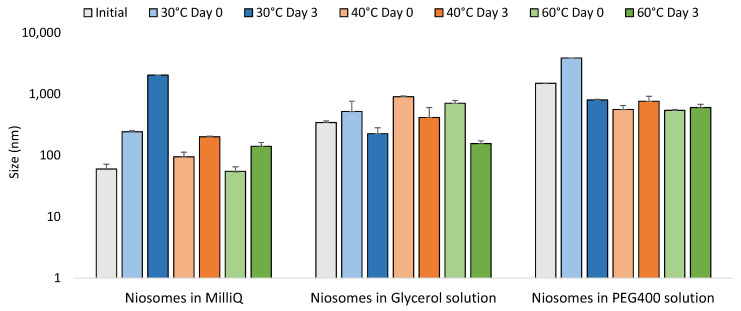
Comparison of size between initial formulations and the formulations at 30 °C, 40 °C, and 60 °C in day 0 and after 3 days.

**Figure 6 pharmaceutics-14-02626-f006:**
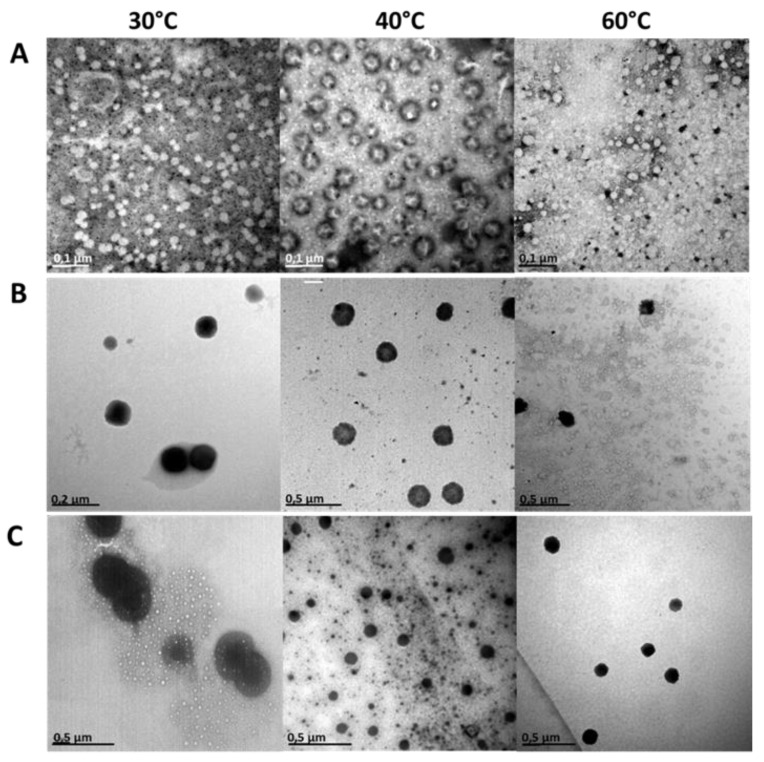
Pictures obtained from HR-TEM of niosomes at different temperatures of 30 °C, 40 °C, and 60 °C. (**A**) Niosomes with rifamycin encapsulated in MilliQ water as the aqueous phase; (**B**) Niosomes with rifamycin S encapsulated in a mixture of MilliQ water and glycerol (60:40 *v*/*v*); (**C**) Niosomes with rifamycin S encapsulated in a mixture of MilliQ water and PEG 400 (44.7:55.3 *v*/*v*).

**Figure 7 pharmaceutics-14-02626-f007:**
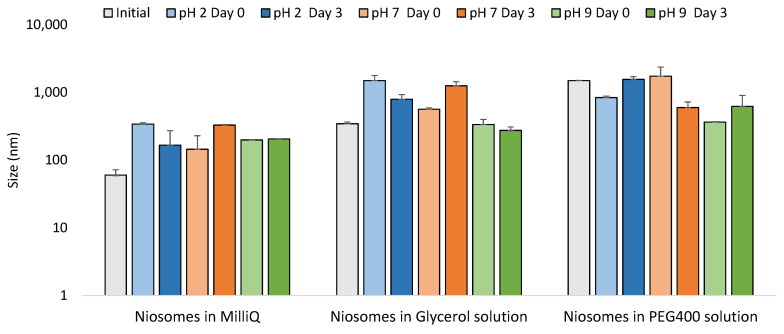
Comparison of size between initial formulations and the formulations at pH 2, pH 7 and pH 9 on day 0 and after 3 days.

**Figure 8 pharmaceutics-14-02626-f008:**
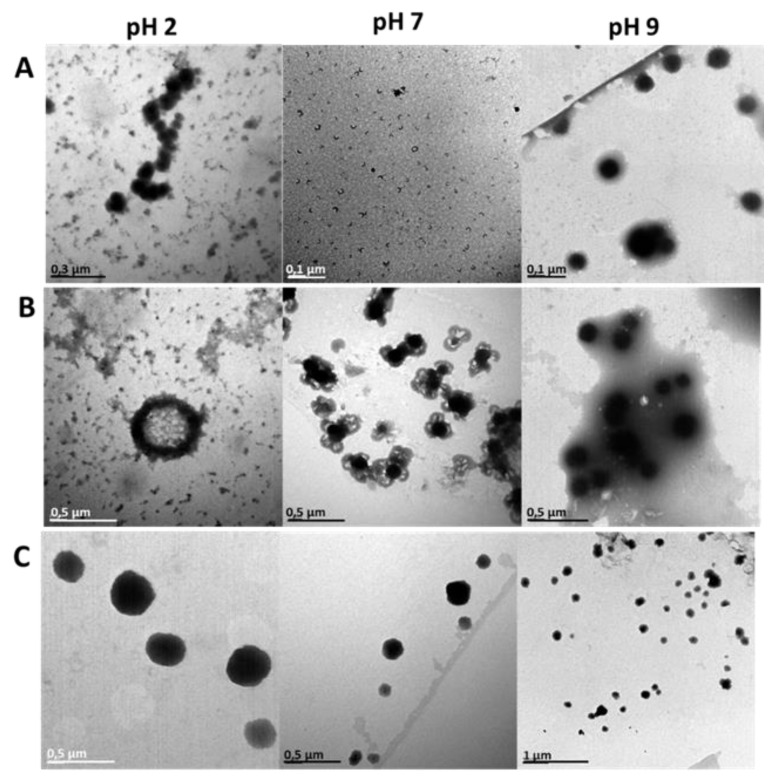
Images obtained from HR-TEM of niosomes at pH 2, 7, and 9. (**A**) Niosomes with rifamycin S encapsulated in MQ water as aqueous phase; (**B**) Niosomes with rifamycin S encapsulated in mixture of MQ water and glycerol (60:40 *v*/*v*); (**C**) Niosomes with rifamycin S encapsulated in mixture of MQ water and PEG 400 (44.7:55.3 *v*/*v*).

**Table 1 pharmaceutics-14-02626-t001:** Mean size and Z-potential values of niosomes formulated with three different aqueous phases.

Formulation	Size (nm)	Z-Potential (mV)
Niosomes in MilliQ	60 ± 12	−31.3 ± 0.5
Niosomes in MilliQ: glycerol	344 ± 21	3.5 ± 0.5
Niosomes in MilliQ: PEG400	1498 ± 11	−1.1 ± 0.7

**Table 2 pharmaceutics-14-02626-t002:** MIC values of niosomes hydrated with glycerol and PEG400 solutions with rifamycin S versus that of free rifamycin S.

Formulations	MIC (µg/mL)
Niosomes in MilliQ: glycerol + rifamycin S	0.025
Niosomes in MilliQ: PEG400 + rifamycin S	0.05
Free rifamycin S	<0.025

**Table 3 pharmaceutics-14-02626-t003:** Physical properties of vesicles hydrated in three aqueous phases at room temperature and heated at different temperatures (30 °C, 40 °C and 60°C), fresh and after 3 days of storage.

		MilliQ	Glycerol Solution	PEG400 Solution
Size (nm)	Fresh	60 ± 12	344 ± 21	1498 ± 11 ^b^
	30 °C	243 ± 10	521 ± 72	3877 ± 11
	40 °C	95 ± 18	907 ± 29	562 ± 88 ^b^
	60 °C	55 ± 10	712 ± 76 ^a^	545 ± 19
	30 °C—3 days	2045 ± 8	227 ± 55	802 ± 24
	40 °C—3 days	202 ± 5	417 ± 66	768 ± 61
	60 °C—3 days	141 ± 22	156 ± 16 ^a^	603 ± 78
Z-pot (mV)	Fresh	−31 ± 1	3.5 ± 0.5	−1.1 ± 0.7
	30 °C	−28.2 ± 0.4	44.0 ± 0.8	9.9 ± 0.2
	40 °C	−27.0 ± 0.8	27.2 ± 0.4	−4.2 ± 0.8
	60 °C	−37.9 ± 0.9	18.3 ± 0.3	−3.3 ± 0.2
	30 °C—3 days	−27.7 ± 0.5	1.5 ± 0.9	−3.3 ± 0.9
	40 °C—3 days	−39 ± 1	−4.6 ± 1.1	−9 ± 1
	60 °C—3 days	−21.3 ± 0.7	−7.2 ± 0.7	35 ± 2
EE (%)	Fresh	76 ± 9	81 ± 2	84 ± 2
	30 °C—3 days	84 ± 5	81 ± 3	72 ± 3
	40 °C—3 days	77 ± 6	88 ± 5	65 ± 2
	60 °C—3 days	64 ± 4	96 ± 9	49 ± 6
LC (%)	Fresh	0.011 ± 0.001	0.012 ± 0.001	0.012 ± 0.001
	30 °C—3 days	0.012 ± 0.001	0.012 ± 0.001	0.010 ± 0.001
	40 °C—3 days	0.011 ± 0.001	0.013 ± 0.001	0.009 ± 0.001
	60 °C—3 days	0.009 ± 0.001	0.013 ± 0.001	0.007 ± 0.001
TSI	30 °C—3 days	30.88	18.79	19.09
	40 °C—3 days	27.48	18.55	11.36
	60 °C—3 days	7.17	2.13	7.54

Measurements with the same superscript letter indicate no significant difference, according to Fisher’s test (*p* < 0.05).

**Table 4 pharmaceutics-14-02626-t004:** Physical properties of vesicles hydrated in three aqueous phases at natural pH and at other pHs (2, 7, 9), fresh and after 3 days under storage.

		MilliQ	Glycerol Solution	PEG400 Solution
Size (nm)	Fresh	60 ± 12 ^a^	344 ± 21 ^b^	1498 ± 11 ^c,d^
	2	340 ± 17	1497 ± 278 ^b^	841 ± 40
	7	166 ± 53	793 ± 54	1553 ± 163 ^c^
	9	145 ± 64 ^a^	565 ± 27	1745 ± 304 ^d^
	2–3 days	330 ± 16	1252 ± 186	598 ± 44
	7–3 days	199 ± 12	336 ± 63	365 ± 14 ^c^
	9–3 days	205 ± 11	274 ± 35	620 ± 81
Z-pot (mV)	Fresh	−31.3.3 ± 0.5	3.5 ± 0.5	−1.14 ± 0.7
	2	13.8 ± 0.8	15 ± 1	19 ± 2
	7	−36 ± 2	10 ± 1	33 ± 3
	9	−38 ± 1	13.1 ± 0.1	21 ± 2
	2–3 days	8.9 ± 0.7	0.4 ± 0.8	−0.1 ± 4.2
	7–3 days	−24.6 ± 0.6	0.3 ± 2	−15 ± 1
	9–3 days	−25.9 ± 0.5	10 ± 1	−8.6 ± 0.9
EE (%)	Fresh	76 ± 9	81 ± 5	84 ± 5
	2	61 ± 5	63 ± 5	49 ± 5
	7	86 ± 3	91 ± 6	85 ± 7
	9	91 ± 9	70 ± 4	93 ± 7
LC (%)	Fresh	0.011 ± 0.001	0.012 ± 0.001	0.012 ± 0.001
	2	0.008 ± 0.001	0.009 ± 0.001	0.010 ± 0.001
	7	0.013 ± 0.001	0.013 ± 0.001	0.012 ± 0.001
	9	0.014 ± 0.001	0.010 ± 0.001	0.014 ± 0.001
TSI	2–3 days	31.43	46.15	18.83
	7–3 days	3.47	8.60	17.32
	9–3 days	3.24	5.39	18.89

Measurements with the same superscript letter indicate no significant difference, according to Fisher’s test (*p* < 0.05).

## Data Availability

Not applicable.

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
