# Peer review of "Stability of Non-Ionic Surfactant Vesicles Loaded with Rifamycin S"

_pharmaceutics, 2022, doi:10.3390/pharmaceutics14122626_

Round 1
Reviewer 1 Report
The article “Stability of non-ionic surfactant vesicles loaded with Rifamycin S” by V. Marchianò and colleagues investigates the conditions for synthesizing niosomes encapsulating rifamicin S and their stability at different temperatures and pH and their antimicrobial activity.
The article is well-organized but several aspects need to be addressed in order to improve the quality of the article.
Control experiments with equal amount of free Rifamycin S must be performed to assess the utility of the niosome formulation.
In order to assess a synergistic effect, combined inhibition by the 2 drugs must be greater than that of each drug alone (e.g. https://doi.org/10.1177/1947601912440575). This has not been tested and quantified but it should (also with the free drugs).
Several editing errors must be corrected, e.g. spaces between numbers and units, liter goes with the capital letter “L” (in uL, mL, etc.), S. Aureus goes in italics.
Several English errors are also present and must be corrected.
Rpm must be converted to either g or rcf.
Why did the authors measure EE in that way? EE is calculated by ((encapsulated drug)/(total drug added)) x100, i.e. ((total drug added- non-encapsulated drug)/(total drug added)) x100.
Figure 5 could be moved to the supplementary information while a graph reporting the change in size as a function of the temperature at the different times (0 and 3 days) and synthetic conditions (milliQ, glycerol, and PEG400) would be more useful in the main article. The same applies to the pH study. In addition, the interpretation of the results doesn’t seem so obvious to me, maybe the use of statistical analysis and/or a software (e.g. Design of Experiments) could help to avoid speculations.
Author Response
Reviewer #1: “The article “Stability of non-ionic surfactant vesicles loaded with Rifamycin S” by V. Marchianò and colleagues investigates the conditions for synthesizing niosomes encapsulating rifamicin S and their stability at different temperatures and pH and their antimicrobial activity. The article is well-organized, but several aspects need to be addressed in order to improve the quality of the article.”
- Control experiments with equal amount of free Rifamycin S must be performed to assess the utility of the niosome formulation.
The control experiment performed with free antiobitic was included in the revised version of the manuscript. MIC obtained in this case was lower than for systems with antiobiotic encapsulated. However, the main purpose of encapsulation is to preserve the antiobitic properties versus external agents, control delivery and reduce the adverse side-effects.
- In order to assess a synergistic effect, combined inhibition by the 2 drugs must be greater than that of each drug alone (e.g.https://doi.org/10.1177/1947601912440575). This has not been tested and quantified but it should (also with the free drugs).
Authors thank the information indicated by the referee. An isoboles method was followed to analyse the combination effect of both compounds (Rifamicyn S and glycerol) according to the method described in the indicated reference. The system indicate a subadditive effect. The manuscript has been modified according to the obtained results.
- Several editing errors must be corrected, e.g. spaces between numbers and units, liter goes with the capital letter “L” (in uL, mL, etc.), S. Aureus goes in italics.
It has been revised and corrected in the revised version of the manuscript. Changes are highlighted in blue.
- Several English errors are also present and must be corrected.
English language has been revised and corrected in the new version of the manuscript. Changes are highlighted in blue.
- Rpm must be converted to either g or rcf.
Thank for the indication. It was a mistake, since the value was already in g. It has been corrected and highlighted in the revised version of the manuscript.
- Why did the authors measure EE in that way? EE is calculated by ((encapsulated drug)/(total drug added)) x100, i.e. ((total drug added- non-encapsulated drug)/(total drug added)) x100.
Thank you very much for the appreciation. We think it is more exactly because some material can be lost during purification (filtration, centrifugation ...) what could lead to overestimated EE values. However, the authors agree that often the suggested method is used in colloidal systems (such as emulsions) as is the case in other research group work (Matos, M., et al. "O/W emulsions stabilized by OSA-modified starch granules versus non-ionic surfactant: Stability, rheological behaviour and resveratrol encapsulation." Journal of Food Engineering 222 (2018): 207-217; Diaz-Ruiz, Rocio, et al. "Enhancing trans-Resveratrol loading capacity by forcing W1/O/W2 emulsions up to its colloidal stability limit." Colloids and Surfaces B: Biointerfaces 193 (2020): 111130.).
- Figure 5 could be moved to the supplementary information while a graph reporting the change in size as a function of the temperature at the different times (0 and 3 days) and synthetic conditions (milliQ, glycerol, and PEG400) would be more useful in the main article. The same applies to the pH study. In addition, the interpretation of the results doesn’t seem so obvious to me, maybe the use of statistical analysis and/or a software (e.g. Design of Experiments) could help to avoid speculations.
New Figures 5 and 7 have been included in the revised version of the manuscript showing mean sizes of all systems at different storage conditions, T and pH, respectively. Particle size distribution has been moved to the supplementary material.
Statistical analysis ANOVA has been performed in order to identify significant differences between obtained values.
Reviewer 2 Report
This is an interesting study about stability of non-ionic surfactant vesicles loaded with Rifamycin S. I recommend it for publication after the points below are well addressed.
1. Line 45-46, 'Nanomaterials are widely used in the therapeutic field for the transport of anticancer and antimicrobial drugs.' Several studies (Journal of Materials Chemistry B 4 (31), 5256-5264; Pharmaceutics 12.2 (2020): 171; Acta Biomaterialia 103 (2020): 223-236) are recommended to be included to support such a claim.
2. Are the lipids and other components stable under high-power sonication?
3. How was the ethanol removed after loading?
4. The drug loading content should be calculated.
5. Figure 2, please show the intensity data.
6. The quality of figure 2,4,5, and 7 should be improved to a publishable level.
7. Formatting issue. Line 98, 'μ l' to 'μ L'. The style of table 1 is not standard. Please check all.
Author Response
Reviewer #2: “This is an interesting study about stability of non-ionic surfactant vesicles loaded with Rifamycin S. I recommend it for publication after the points below are well addressed”.
- Line 45-46, 'Nanomaterials are widely used in the therapeutic field for the transport of anticancer and antimicrobial drugs.' Several studies (Journal of Materials Chemistry B 4 (31), 5256-5264; Pharmaceutics 12.2 (2020): 171; Acta Biomaterialia 103 (2020): 223-236) are recommended to be included to support such a claim.
Authors thank the referee for the valuable works that has suggest. They have been included in the revised version of the manuscript.
- Are the lipids and other components stable under high-power sonication?
Conditions used for the sonication were chosen considering the stability of the components of vesicles checking previous works (for example: Pando, D., et al. "Preparation and characterization of niosomes containing resveratrol." Journal of Food Engineering 117.2 (2013): 227-234) and the sonication has been carried out continuous mode and a cooling system was used to avoid increase of sample temperature.
The authors recognize that the high-power sonication method to obtain a homogeneous vesicle size has disadvantages, such as the probe can release titanium metal residues (Ge, Xuemei, et al. "Advances of non-ionic surfactant vesicles (niosomes) and their application in drug delivery." Pharmaceutics 11.2 (2019): 55.) and that it can also be used as a method for breaking colloidal systems (Salabat, Alireza, and Shima Soleimani. "Ultrasonic irradiation and solvent effects on destabilization of colloidal suspensions of platinum nanoparticles." Particuology 17 (2014): 145-148).
- How was the ethanol removed after loading?
The ethanol solution (containing the surfactants and antibiotic) is placed in contact with a high-pressure nitrogen flow which allows to evaporate the ethanol from the solution creating a thin film on the bottom of the flask. It is clearer indicated in the revised version of Figure 1.
- The drug loading content should be calculated.
Authors thank the referee for the suggestion and drug loading was calculated in the revised version of the manuscript and included in Figure 4 and Tables 3 and 4.
- Figure 2, please show the intensity data.
Figure 2 was changed in the revised manuscript showing the results related to the intensity values.
- The quality of figure 2,4,5, and 7 should be improved to a publishable level.
The authors thanks for the suggestion and the quality of the figures has been improved in the revised version of the manuscript. High quality figures file has been uploaded to the system separately.
- Formatting issue. Line 98, 'μ l' to 'μ L'. The style of table 1 is not standard. Please check all.
The manuscript has been revised and modified according to standard symbols.

Round 2
Reviewer 1 Report
The authors addressed most of the issues raised and performed few additional improvements. In my opinion, the article is now suitable for publication in Pharmaceutics.